# Mechanisms and Strategies to Overcome PD-1/PD-L1 Blockade Resistance in Triple-Negative Breast Cancer

**DOI:** 10.3390/cancers15010104

**Published:** 2022-12-23

**Authors:** Xingyu Chen, Lixiang Feng, Yujing Huang, Yi Wu, Na Xie

**Affiliations:** 1West China School of Basic Medical Sciences and Forensic Medicine, Sichuan University, Chengdu 610041, China; 2State Key Laboratory of Biotherapy and Cancer Center, West China Hospital, Collaborative Innovation Center for Biotherapy, Chengdu 610041, China

**Keywords:** triple-negative breast cancer, immune checkpoint inhibitor, PD-1/PD-L1, resistance, combination therapy

## Abstract

**Simple Summary:**

Breast cancer is the most prevalent malignancy in women. With the improvement of medical treatment, breast cancer has become one of the solid tumors with the best curative effect. However, triple-negative breast cancer is not sensitive to conventional treatment due to its high invasiveness, resulting in a poorer prognosis than other types. The antibodies of programmed death receptor 1 and its ligands represent a new option as immunotherapy for patients with triple negative breast cancer. However, some recent clinical data suggest that a large proportion of patients exhibit primary or acquired resistance to treatment with programmed death receptor antibodies. In this review, we discuss the mechanisms that lead to resistance and also summarize potential strategies to overcome the resistance, improving the therapeutic efficacy of programmed death receptor 1 and its ligand-based antibodies in triple negative breast cancer.

**Abstract:**

Triple-negative breast cancer (TNBC) is characterized by a high rate of systemic metastasis, insensitivity to conventional treatment and susceptibility to drug resistance, resulting in a poor patient prognosis. The immune checkpoint inhibitors (ICIs) represented by antibodies of programmed death receptor 1 (PD-1) and programmed death receptor ligand 1 (PD-L1) have provided new therapeutic options for TNBC. However, the efficacy of PD-1/PD-L1 blockade monotherapy is suboptimal immune response, which may be caused by reduced antigen presentation, immunosuppressive tumor microenvironment, interplay with other immune checkpoints and aberrant activation of oncological signaling in tumor cells. Therefore, to improve the sensitivity of TNBC to ICIs, suitable patients are selected based on reliable predictive markers and treated with a combination of ICIs with other therapies such as chemotherapy, radiotherapy, targeted therapy, oncologic virus and neoantigen-based therapies. This review discusses the current mechanisms underlying the resistance of TNBC to PD-1/PD-L1 inhibitors, the potential biomarkers for predicting the efficacy of anti-PD-1/PD-L1 immunotherapy and recent advances in the combination therapies to increase response rates, the depth of remission and the durability of the benefit of TNBC to ICIs.

## 1. Introduction

According to the latest data from the International Agency for Research on Cancer of the World Health Organization, breast cancer has replaced lung cancer as the most prevalent malignancy worldwide [1]. Breast cancer is classified into four molecular types based on the expression of estrogen receptor (ER), progesterone receptor (PR), human epidermal growth factor receptor-2 (HER-2) and Ki-67 in breast cancer. Among them, triple negative breast cancer (TNBC) is characterized by negative ER, PR and HER-2, accounting for 10–20% [2]. Compared with other subtypes of breast cancer, TNBC is more aggressive, with a higher risk of recurrence and a poorer prognosis. Traditional breast cancer treatments include chemotherapy, endocrine therapy and targeted therapy. However, TNBC is unsuitable for endocrine and targeted therapy due to the lack of corresponding targets [3]. TNBC also has an unsatisfactory response to chemotherapy. Patients with advanced TNBC experience early onset of drug resistance during chemotherapy, with a median progression-free survival (PFS) of only 3–6 months and a median overall survival (OS) of only 10–13 months [4]. In recent years, the emergence of immune checkpoint inhibitors (ICIs) is radically altering our conceptions of cancer treatment. Targeting the PD-1/PD-L1 axis with ICIs yields significant anti-tumor activity and may provide long-term survival benefits, particularly for patients with TNBC [5,6]. For PD-L1-positive TNBC patients, treatment with atezolizumab in combination with nab-paclitaxel prolonged median overall survival by 10 months compared to chemotherapy alone [7].

Programmed death receptor 1 (PD-1) is a member of the CD28 superfamily that is expressed mainly in activated T-lymphocytes and myeloid cells, functioning as a crucial immunosuppressive molecule [8,9]. The PD-1 mainly consists of an extracellular immunoglobulin variable region (IgV), a hydrophobic transmembrane region and an intracellular region [10]. The tail of the intracellular region has an immunoreceptor tyrosine-based inhibitory motif (ITIM) and an immunoreceptor tyrosine-based switch motif (ITSM). PD-1 is an essential immune checkpoint receptor for activated T cells and plays a critical role in immunosuppression control. Binding to its ligand programmed death ligand-1 (PD-L1) induces the phosphorylation of tyrosine in ITSM of PD-1, which dephosphorylates downstream protein kinases Syk and PI3K, inhibiting the transcription and translation of genes and cellular factors required for T cell activation (Figure 1) [11,12]. Tumor cells can inhibit the killing function of T cells by high expression of PD-L1, thus contributing to immune escape. Tumor cells upregulate PD-L1 expression mainly through the following pathways: activation of EGFR, MAPK or PI3K-Akt pathway [13,14,15]; high expression of STAT3 and HIF-1 [16] alteration of PD-L1 at the genetic level, for example, the PD-L1 3′ untranslated region (3′ UTR) plays a negative regulatory role in PD-L1 expression, and loss of this gene fragment due to different structural variants correlates with increased PD-L1 expression in tumor cells [17,18]; and microRNA-based control of PD-L1 expression, for example, miR-513 overexpression can block IFN-γ-induced PD-L1 expression in bile duct cells [19,20]. The upregulated PD-L1 binds to PD-1 on the surface of tumor-specific CD8^+^ T cells, suppressing an anti-tumor immune response. In addition, elevated inflammatory factors such as IFN-γ in the tumor microenvironment (TME) can also induce PD-L1 and PD-L2 expression, resulting in “adaptive immune resistance” [21].

Immune checkpoint inhibitors (ICIs) have been developed to boost the immune killing of tumors by restoring the function of tumor infiltrating CD8^+^ T cells. ICIs, including PD-1 monoclonal antibody (pembrolizumab), PD-L1 monoclonal antibody (atezolizumab) and CTLA-4 monoclonal antibody (durvalumab), are commonly used in clinical [22,23,24]. These ICIs can restore or improve the patient’s anti-tumor immunity to kill tumor cells by overcoming tumor cell-mediated immune cell dysfunction. In recent years, ICI-based immunotherapy has made breakthroughs in treating melanoma and non-small cell lung cancer. In contrast, breast cancer immunotherapy has remained stagnant because breast cancer was previously considered an immunologically “cold tumor” with low immunogenicity [25,26]. TNBC is the most immunogenic subtype of breast cancer with higher levels of PD-L1 expression and tumor infiltrating lymphocytes (TILs) than other subtypes, suggesting that it is more likely to benefit from treatment with ICIs [27]. However, the limited effectiveness of TNBC to ICIs due to heterogeneity and immunosuppressive microenvironment must be overcome [7,28]. Therefore, it is vital to uncover the underlying mechanism of therapeutic resistance and partial hypo-responsiveness to effectively improve the clinical response rate of cancer patients to ICI treatment. This review summarizes the mechanisms of therapeutic resistance to ICIs in TNBC patients. It proposes potential solutions for overcoming ICI resistance, thereby providing theoretical support for enhancing its clinical anti-TNBC efficacy.

## 2. Resistance Mechanism of PD-1/PD-L1 Inhibit Therapies

Although ICI therapy has a successful anti-tumor effect in some patients, therapeutic resistance prevents ICIs from being used more widely in the clinical setting. ICI resistance’s complicated and varied mechanisms fall into the three following categories: primary, adaptive and acquired [29]. In contrast to primary resistance, in which tumors initially do not respond to ICIs, acquired resistance occurs after effective ICI treatment, leading to cancer progression or recurrence. Adaptive resistance refers to a tumor’s ability to evade immune attack despite the immune system can recognize it [30]. In this section, we discuss the current mechanisms of ICI resistance from the perspectives of antigen presentation, an immunosuppressive TME, the interplay of multiple immune checkpoints and the abnormal signaling pathways in tumors. 

### 2.1. Disturbed Presentation of Tumor-Specific Antigens

The efficacy of ICIs on tumor cells depends on the killing effects of cytotoxic T lymphocytes (CTLs), which can target tumor-specific antigens (TSAs). The specific load and expression pattern of TSAs associated with the TMBs contribute to the different immunogenicity of tumors. Tumors with low or down-regulated TSAs can escape the immune system surveillance and develop tolerance in the body. Due to the lack of highly immunogenic TSAs to activate the TILs, low-immunogenic malignancies, like pancreatic cancer, have an inefficacy response to PD-1/PD-L1 antibodies [21,31]. However, the tumor cells with highly immunogenic antigens can selectively hide the antigens by reducing gene expression or deleting mutant alleles to avoid T cell-dependent cancer immunoediting [32,33,34]. Breast cancer has a substantially lower median TMB than melanoma and lung cancer, indicating that it is generally less immunogenic. The immunogenicity also varies between subtypes of breast cancer. In a study of 762 primary tumors, the mean mutational burden identified in ER^−^ tumors was more significant than in ER^+^ ones [35,36]. The whole exome sequencing (WES) on 3969 patients with primary or metastatic breast cancer revealed that only 5% (198 cases) had hypermutated genes. These hypermutated tumor subgroups, including TNBC, exhibited increased neoantigen burden and cytolytic activity [37]. Consistent with the reported disparities in mutational burden, a higher response rate of anti-PD-1 therapy in PD-L1-positive tumors is observed in TNBC than in ER^+^ breast cancer (18.5% vs. 12%) [38]. Although TNBC is the most immunogenic subtype of breast cancer, its response rate to ICIs remains poor. The immunogenicity may decrease during tumor progression due to reduced antigen expression and impaired antigen presentation [39].

Tumors can avoid T cell cytotoxicity by downregulating the antigen presentation by major histocompatibility complex (MHC) molecules [40,41]. In a study of 117 primary breast cancer tissue sections, more than half of the cases (19/32, 59%) in TNBC exhibited MHC-I deficiency, allowing tumor cells to evade the cytotoxic T cell-mediated immune response [42]. H3K27me3 alterations in TNBC reduce MHC-I transcription, which can be restored by epigenetic modulators, such as chidamide and pharmacological inhibitors of PRC2 subunits EZH2 or EED [43,44]. Moreover, the elevated MYC and MAL2 in TNBCs drive immune evasion by reducing MHC-I expression, resulting in resistance to ICI therapy [45,46]. In addition to MHC-I, short MHC-II antigen presentation contributes to ICI resistance. MHC-II expression in tumor cells was tightly related to increased TILs and interferon signaling in TNBC [47]. Compared to MHC-II-negative patients, pembrolizumab-treated TNBC patients with MHC-II expression had a more excellent pCR and better prognosis [48]. However, MHC-II on breast cancer cells provides selection pressure for LAG-3^+^ and FCRL6^+^ TILs, which antagonizes MHC-II expression and suppresses the antigen presentation, promoting adaptive resistance to anti-PD-1 treatment [49]. The alteration of other tumor antigen processing and presentation molecules, such as LMP and transporter for antigen presentation (TAP), can also result in ICI resistance [41,50].

Antigen presentation can also be impaired by the immunosuppressive TME, which results in resistance to ICIs. The aberrant differentiation of myeloid cells caused by the IL-10 released by TNBC reduces the number of dendritic cells (DCs) and increases the expression of PD-L1. Simultaneously, CD80 and CD86 cannot be activated due to the absence or low expression of co-stimulatory molecules in peripheral immature DCs, resulting in a decreased proportion of normal mature DCs and an increased immature DCs in peripheral blood. After infiltration into tumors, these immature DCs cannot generate an immune response due to their impaired antigen presentation function and inability to activate T cells, resulting in immune escape and resistance to ICIs [51,52,53].

### 2.2. Immunosuppressive TME

Numerous immune cells, stromal cells and cytokines are present in the TME of TNBC, which may affect the response of tumor cells to immunotherapy. Tumor-derived cytokines recruit more immunosuppressive cells into TME, which affect the efficacy of anti-PD-1/PD-L1 by suppressing T cell activity. Thus, the alteration of TME is a vital mechanism causing immunotherapy resistance [54].

#### 2.2.1. Immunosuppressive Cells

The function of anti-tumor T cells is suppressed by various immunosuppressive cells found in the TME, such as T_reg_ cells, marrow-derived suppressor cells (MDSCs), and tumor-associated macrophages (TAM) (Figure 2). T_reg_ cells, characterized by the expression of Foxp3, CD25 and CD4, suppress the immune response of other immune cells, functioning as the primary controller of self-tolerance [55]. It has been demonstrated that a high abundance of T_reg_ is significantly associated with a high level of PD-L1 in TNBC and results in low pCR with ICI treatment [56]. Patients with TNBC who have T_reg_ cells in their TME have a worse prognosis, lower relapse-free survival rates and worse rates of overall survival [57,58]. T_reg_ cells inhibit the proliferation of effector T cells by consuming the IL-2 and secreting immunosuppressive molecules like TGF-β, IL-35 and IL-10 [59,60]. The CTLA-4 on T_reg_ cells binds to co-stimulatory molecules CD80 and CD86 on APCs to inhibit the secondary signaling, suppressing antigen presentation [61]. Treg can also directly destroy effector T cells and APCs by releasing perforin and granzyme [62]. Therefore, the Treg infiltration in TNBC inhibits T cells’ cell proliferation and function, leading to therapeutic resistance.

MDSCs, consisting of early myeloid cells, macrophages, immature granulocytes and DCs at different stages of differentiation, are a heterogeneous group of immature bone marrow cells with the capacity to control local immune responses [63]. In patients with TNBC, MDSCs may play a significant role as negative modulators of anti-tumor immunity [64]. TNBC patients have higher levels of MDSCs recruited by chemokines and cytokines produced from tumor cells, including IL-34, CXCL2 and CCL22 [65,66,67]. The recruitment of MDSCs at tumors hinders CTL infiltration and challenges the anti-tumor potentials of T cell-based immunotherapies. MDSCs result in resistance by inhibiting both the innate and adaptive immune response [68,69]. Additionally, MDSCs play a nonimmunologic role in promoting tumor progression by releasing prometastatic factors, including MMP9 and chitinase 3-like 1 [65]. In PD-1 monoclonal antibody-treated mice, reducing the MDSC recruitment by PI3K inhibitors achieves a combined inhibitory effect with PD-1 monoclonal antibody [70,71].

TAMs are differentiated monocytes recruited into the tumors by chemokines like CSF1 and CCL2 [72]. While M1 type macrophages destroy tumor cells and prevent pathogen infection, M2 type macrophages primarily promote tumor growth, invasion and metastasis [73]. A high density of M2 macrophages in TNBC promotes cancer cell proliferation and is associated with a greater risk of metastasis and a worse prognosis [73]. TNBC cells with BRCA1-IRIS-overexpression (IRISOE) can release high quantities of GM-CSF, which attracts macrophages to tumor cells and polarizes them to protumor M2 TAMs. This interplay of IRISOE cells and macrophages results in an immunosuppressive milieu in TNBC tumors that is conducive to the development of immune-evading TNBC [74]. M2 type TAM-mediated progression of TNBC can be stopped by blocking MAPK signaling with MEK inhibitors [75]. Additionally, TAMs have been demonstrated to directly and indirectly regulate the effect of anti-PD-1/PD-L1 on tumor cells by regulating the PD-1/PD-L1 expression, reducing the effector function of PD-1^+^ TILs, promoting PD-1^+^ Treg development and activity [76]. CSF-1R inhibitors are available to block macrophage CSF-1R and reduce the number of M2-type macrophages, ultimately increasing the response of tumor cells to PD-1 antibodies [77].

#### 2.2.2. Cytokines

Chemokines and cytokines are another class of modulators in immunotherapy resistance via recruiting the immunosuppressive and regulating the expression of PD-1/PDL-1 in the TME. Elevated TGF-β results in the poor prognosis of TNBC by inducing EMT in tumor cells, recruiting immunosuppressive cells and suppressing CD8^+^ T cell function [78]. TGF-β has also been found to induce the expression of PD-L1 to promote tumor escape [76]. The TGF-β inhibitor tranilast enhances the anti-tumor effect of immunotherapy by improving the hypoxic environment [79]. The pro-inflammatory cytokine IL-6 released by TNBC cells induces the production of CCL5 and VEGF in lymphatic endothelial cells (LEC), promoting TNBC lymph node metastasis and angiogenesis [80,81,82]. Blocking the IL-6/CCR5 or VEGF signaling can prevent TNBC from metastasis and enhance the inhibitory effect of ICIs on TNBC by upregulating PD-L1 [81]. The levels of tumor-derived IL-18 is significantly correlated with poor survival in TNBC patients. IL-18 in TME of TNBC increases the number of immunosuppressive NK cells and induces PD-1 expression in NK cell subsets, resulting in resistance to ICIs [83,84]. Additionally, the increased expression of guanylate binding protein 5 (GBP5) and deletion of the tumor suppressor transcription factor Elf5 activate the IFN-γ signaling pathway (Figure 3) and promote PD-L1 expression, causing immunotherapy resistance in TNBC [85,86].

#### 2.2.3. T Cell Exhaustion

Additional factors in TME like hypoxia can cause T cell exhaustion and increase the resistance to immunotherapy (Figure 4). During prolonged exposure to cognate antigens, the upregulated PD-1 expression on T cell surfaces leads to T cell exhaustion, a phenotype characterized by loss of proliferation and cytolytic function, followed by deficiencies in cytokine production. Hypoxia causes dysfunction and terminal exhaustion of human T cells via the epigenetic suppression of immune effector genes [87,88]. The exhausted CD8^+^ T cells with intermediate PD-1 expression can be restored to viability by blocking the PD-1 pathway. However, anti-PD-1 therapy cannot reverse the dysfunctional state of T cells with a high level of PD-1 [89,90]. Reversing the T cell dysfunction using HIF1α/HDAC1/EZH2 inhibitors can overcome the resistance to PD-1 blockade [87].

### 2.3. Compensatory Upregulation of Alternative Immune Checkpoints

Along with PD-1, a variety of immune inhibitory checkpoints, including TIM-3, LAG3, and T cell immune globulin and ITIM structure domain proteins (TIGIT), are highly expressed and linked to T cell function, which also affect the effectiveness of ICIs (Figure 5) [91,92,93].

An immunosuppressive checkpoint protein known as TIM3 (CD366, HAVCR2) is expressed on the surface of activated T cells, NK cells, and monocytes. TIM-3 inhibits cytotoxic T cell activity upon binding to ligands including phosphatidylserine, CEACAM-1, galactin-9, thereby suppressing anti-tumor immunity and promoting tumor escape, and these ligands are members of the h-galactoside-binding protein family that are overexpressed by TNBC cells [94,95,96,97]. In the presence of TNBC cells (MDA-MB-231 and MDA-MB-468), the use of anti-PD-1 monoclonal antibodies causes a compensatory increase in TIM-3 on the surface of CD4^+^ T cells, resulting in a suppressive signal and leads to effector T cell depletion [98]. Early clinical trials using anti-TIM-3 have described an overall acceptable safety profile and initial indications of anticancer activity [99]. Clinical trials are currently investigating whether inhibitors of TIM-3 combined with anti-PD-1/PD-L1 inhibitors can improve efficacy while reducing side effects [28].

The LAG-3 (CD223) is expressed after antigenic stimulation of T cells to prevent their overactivation, maintaining autoimmune tolerance [100]. The persistent LAG-3 expression in response to prolonged antigenic stimulation is associated with exhausted CD8^+^ TILs with a decrease in cytokine secretion and cytolytic activity [101]. A significant positive correlation has been found between the expression of LAG-3 and PD-L1 in TNBC patients [102]. Moreover, blocking PD-1 or PD-L1 in TNBC results in compensatory upregulation of LAG-3 in CD4^+^ T cells [98,103]. A synergistic blockade of PD-1 and LAG-3 in preclinical mouse models exhibits good responsiveness [101]. In a clinical trial, the combination of the anti-LAG-3 antibody leramilimab, the anti-PD-1 antibody spartalizumab and carboplatin had the best rate of remission in patients with advanced TNBC (ORR 32.4%), albeit with successively increased side effects [104]. Therefore, dual inhibition of LAG-3 and PD-1 is feasible for the combination treatment of TNBC in the future.

High levels of TIGIT and its ligand PVR (poliovirus receptor, CD155) are significantly associated with low overall survival and recurrence-free survival of patients with breast cancers [105]. TIGIT is a type I transmembrane protein with an Ig-like variable extracellular structural domain expressed on memory T cells, regulatory T cells and NK cells [106,107,108]. After binding to PVR, TIGIT inhibits immune response through its cytoplasmic immunoglobulin tail tyrosine-like (ITT) phosphorylation motif and ITIM, resulting in resistance in ICI therapy [109]. Dual blocking of TIGIT and PD-1 inhibits tumor growth in a mouse model of breast cancer [110].

### 2.4. Abnormal Signaling Transduction in Tumor Cells

Aberrant oncogenic signaling pathways have been found to have a substantial impact on both the stimulation of immunological escape in TNBC cells and the formation of immunosuppressive TME [111]. The activation of the MAPK signaling pathway in TNBC induces the production of VEGF and IL-8, which inhibit T cell recruitment and activity [112,113]. The Ras/MAPK pathway can also suppress antigen presentation by inhibiting MHC molecule expression, which helps TNBC cells escape the immune system. A stronger anti-tumor immune response has also been observed in animal models of breast cancer when MEK inhibitors are combined with PD-1/PD-L1 antibodies [113]. In TNBC patients treated with anti-PD-1/PD-L1, alterations in PTEN have been found to be significantly associated with worse ORR and shorter PFS and OS [114]. PTEN gene deletion is linked to reduced T cell infiltration at the tumor site, reduced T cell expansion after tumor resection and poorer efficacy of PD-1 inhibitor therapy in patients with malignant melanoma [115]. Similarly, according to the Cancer Genome Atlas (TCGA) data, 35% of basal-like tumors (mostly TNBC) exhibit PTEN gene deletion, which is associated with activated PI3K signaling, decreased expression of INF-γ and granzyme B, decreased infiltration of CD8^+^ T cell, and also upregulated PD-L1 expression [116,117,118]. The WNT signaling pathway is another potential oncogenic pathway that modulates the immune response. Tumors with high β-catenin expression have low numbers of CD103^+^ DCs in the TME, resulting in impaired antigen delivery and presentation, which in turn affects T cell infiltration and immune response in the microenvironment [119]. TNBC stem cells also constitutively upregulate PD-L1 through the activated WNT signaling pathway [120].

## 3. Biomarkers for Predicting the Efficacy of Anti-PD-1/PD-L1 Immunotherapy

PD-1/PD-L1 antibodies could restore pre-existing tumor-specific T cells by reducing the inhibitory effect of an active PD-1/PD-L1 axis. Therefore, the efficacy of PD-1/PD-L1 therapy is dependent upon the activity of tumor-reactive T lymphocytes in eradicating tumors, which is associated with the levels of PD-L1 expression, tumor mutational burden (TMB) and microsatellite instability (MSI), TILs and mismatch repair deficiencies. Using these indicators, it is critical to select TNBC patients who are more likely to respond to ICIs.

### 3.1. PD-L1 Expression Level

The PD-L1 expression levels on tumor cells and immune cells are the primary determinant of ICI response. Therefore, it is a significant biomarker for predicting the efficacy of PD-1/PD-L1 inhibitors in solid tumors, including non-small cell lung cancer (NSCLC), gastric cancer, esophageal cancer, uroepithelial cancer and cervical cancer [121,122]. A study on 654 different tumor specimens found that TNBC had higher PD-L1 expression than other breast cancer subtypes. Similarly, TNBC accounted for 59% of breast tumors with high levels of PD-L1 expression, supporting the feasibility of PD-L1 serving as a biomarker for predicting the ICI efficacy in TNBC [122,123,124]. TNBC patients with high PD-L1 expression are more likely to benefit from PD-1/PD-L1 blockade therapy [125]. Large clinical trials have confirmed that PD-L1-positive patients benefit from ICI therapy using several criteria, such as the combined positive score (CPS) in KEYNOTE-355 and the percentage of TILs in IMpassion130 [126,127]. However, some PD-L1-negative patients still respond to ICIs, which complicates the issue of PD-L1 as an exclusionary predictive biomarker [128]. This inconsistency of results can be partially attributed to the PD-L1 expression on non-tumor cells in the TME, which can promote immune escape of tumors [129]. Additionally, patients with negative biomarkers respond when intersecting biological pathways are activated without the presence of a specific biomarker [122]. In summary, although it is not the only consideration when deciding whether to administer an ICI, PD-L1 expression is one of the important predictors of whether patients will benefit from them.

### 3.2. TILs

TNBC subtype is associated with the highest TILs levels. TILs are the lymphocytes that aggregate in the tumor tissue or surrounding stroma, which might predict the efficacy of immunotherapy in epithelial tumors [130,131]. It has been shown that TIL levels were higher in TNBC (30%) than that in HER2-positive breast cancer (19%) and luminal breast cancer (13%). Consistently, TNBC usually has greater genetic instability, leading to high TMB and a robust anti-tumor immune response, implying that this subtype is more immunogenic [132]. Rich TILs were highly related to improved survival outcomes in early TNBC, making them a powerful immunotherapy prognostic factor for this subtype [133]. In the IMassion130 clinical study treating advanced or metastatic TNBC with atezolizumab and nab-paclitaxel, PFS and OS were significantly higher in patients with both abundant TILs and high PD-L1 expression [134]. Similarly, high levels of TILs and PD-L1 are also positively associated with pathologic complete remission and overall remission rates in TNBC patients receiving pembrolizumab and chemotherapy in the KEYNOTE-173 study [135]. Collectively, the TILs constitute a potential criterion for selecting patients who will benefit from ICI therapy.

### 3.3. TMB and MSI

DNA damage is a common phenomenon during biological evolution, which induces mutations that drive carcinogenesis and cancer recurrence. The total amount of somatic mutations, termed TMB, is another predictor of the anti-tumor efficacy of ICIs [136]. Tumor cells with non-synonymous mutations can generate neoantigens that are recognized by T cells as non-self antigenic epitopes, initiating the immunological clearance of tumor cells. The increased production of neoantigens due to a higher level of TMB elicits a stronger tumor-specific immune response [137]. It has been demonstrated that patients with high levels of TMB in melanoma, lung cancer and colorectal cancer respond better to ICIs, increasing their chances of survival [21,138,139,140]. Accordingly, breast cancer patients with high TMB (≥10 mut/Mb) may benefit from ICIs regardless of the mutational process [37,141]. A phase II multicenter trial of nivolumab plus ipilimumab for metastatic HER2-negative breast cancer with high TMB has been conducted to elucidate the impact of high TMB [114]. The median TMB for TNBC was 2.630 mut/Mb, with a prevalence of 5% mutation. Clinical studies suggest that TNBC patients with a high TMB have a longer PSF after being treated with ICIs. Therefore, TMB is one feature of TNBC that can predict response to ICI therapy.

MSI, a unique molecular alteration and hyper-mutable phenotype, is the result of a dysfunctional DNA mismatch repair (MMR) system in cancers. MSI-high (MSI-H) also elevates the mutation burden of tumor-associated genes and the generation of neoantigens, boosting anti-tumor immunity [142,143,144]. However, according to a study in 12,821 different tumor samples from The Cancer Genome Atlas, MMR deficiency was less prevalent in breast cancer with only 1.53% of MSI rates [145]. Furthermore, the expression of the MMR protein MLH1 in TNBC was negatively correlated with PD-L1 expression in stromal immune cells [146]. The higher amounts of tumor-specific antigens induced by MMR gene mutations, which in turn increases the anti-tumor immune response, may account for the greater therapeutic effect of ICIs in MMR-deficient tumors compared to non-MMR-deficient cancers [147]. Similar to this, higher TMB levels lead to increased neoantigen production in TNBC, which may make PD-1/PD-L1 inhibitors more effective [148]. Therefore, MSI might be combined with TMB as a candidate predictor for ICI treatment in TNBC patients [149].

### 3.4. Driver Gene Mutation

An increasing number of studies have shown that the gain-of-function of oncogenes suppressed anti-tumor immune response by regulating the expression of immune checkpoints. TNBCs with aberrant activation of MYC are resistant to ICI therapy [45]. MYC increases PD-L1 and CD47 expression by directly binding to their promoters [150,151,152]. Mucin 1 (MUC1) elevates PD-L1 transcription by recruiting MYC and NF-κB to the PD-L1 promoter, contributing to immune escape in TNBC [153]. Nuclear AURKA induces PD-L1 expression via an MYC-dependent pathway to mediate an immune evasion of TNBC, while the downregulation of AURKA leads to increased CD8^+^ T cell infiltration and activation in vivo [154]. Although MYC promotes PD-L1 expression that seems to be a beneficial factor for ICI treatment, it also plays a key role in the regulation of energy metabolism, invasion and angiogenesis, thereby promoting tumor progression [155]. Moreover, it has been observed that THZ1, an inhibitor of the CDK7-p38α-MYC axis, can recruit CD8^+^ T cells to enhance the anti-PD-1 therapeutic effect in a Lewis murine lung cancer model [156]. Therefore, the aberrant activation of oncogenic MYC causes a poor response to ICIs, providing a predictor and target for ICIs in TNBC.

Additionally, mutations in tumor suppressor genes (TSGs) are also related to the immune response of TNBC. A recent genetic study of TNBC patients revealed that those with mutant TP53 and wild-type PIK3CA have higher levels of immune suppressor cells and molecules in their TME. It is expected that patients with this genotype will respond better to immunotherapy [157]. Patients with PD-1 blockade-resistant TNBC have elevated levels of oncogenic LINK-A, which facilitates the K48-polyubiquitination-mediated degradation of the antigen peptide-loading complex (PLC) and intrinsic tumor suppressors Rb and p53, leading to reduced antigen presentation and tumor-specific immune response [158]. Another study employing next-generation sequencing (NGS) on nine patients with primary TNBC has found that patients with heterozygous loss of PTEN have higher PD-L1 expression levels in TILs, suggesting that patients with PTEN mutations may respond better to ICIs [159]. Additionally, the expression of PPP2R2B, a powerful tumor suppressor that plays an important role in the anti-tumor immune response, is significantly downregulated in TNBC tissues compared to normal breast tissues, along with a suppressed T cell receptor signaling pathway, antigen processing and presentation signaling pathway. The involvement of downregulated PPP2R2B in immune evasion renders it a promising predictor for predicting immunotherapeutic response and guiding the treatment of TNBC [160].

### 3.5. TNBC Microenvironment Phenotypes

Each TNBC molecular subtype exhibits distinct TME profiles associated with multiple cell types, including fibroblasts, adipose and immune-inflammatory cells, and blood and lymphatic vascular networks [161]. The TNBC microenvironment phenotypes were classified into three heterogeneous clusters, including the “immune-desert” cluster with low immune cell infiltration, the “innate immune-inactivated” cluster with resting innate immune cells and nonimmune stromal cells infiltration, and the “immune-inflamed” cluster, with abundant adaptive and innate immune cells infiltration. These microenvironment clusters had significant prognostic efficacy [152]. Quiescent cancer cells (QCCs) in TNBC constitute immunotherapy-resistant reservoirs by orchestrating a local hypoxic immune-suppressive milieu that blocks T cell function. The microenvironment of HIF-α1-expressing tumor cells was similar to that of QCC, and the TME after knockout was rich in a large number of killer T cells, suggesting that the hypoxic environment caused by HIF-α1 may affect the efficacy of immunotherapy treatment results, which must be taken into account when predicting treatment outcomes [162]. BRCA1-IRIS-overexpressing (IRISOE) TNBC cells secrete high levels of GM-CSF in a HIF-α1- and NF-κB-dependent manner of recruiting macrophages to tumors. The recruited tumor-associated macrophages (TAMs) derived from peripheral blood monocytes in the TNBC microenvironment promote tumor growth and progression by directly and indirectly modulating PD-1/PD-L1 expression. Accordingly, IRISOE TNBC tumors had significantly few CD8^+^/PD-1^+^ cytotoxic T cells and more CD25^+^/FOXP3^+^ regulatory T cells [74,76]. It has also been found that baseline CD4^+^ levels in the peripheral blood of TNBC patients correlate significantly with PFS and OS, especially in those receiving immunotherapy in combination with chemotherapy. TNBC patients with higher CD4^+^/CD8^+^ ratios have better treatment responses [163].

## 4. Therapeutic Strategies for Immune Checkpoint PD-1/PD-L1 Inhibitor Resistance

As mentioned above, the effectiveness of PD-1/PD-L1 inhibitors alone is only observed in 10%–30% of tumor patients. The efficacy of the same therapy can vary greatly across patients, between different tumor sites in the same patient, and even between different regions of the same tumors due to the high heterogeneity of TNBC. Additionally, immune resistance also contributes to the inadequate response to ICIs. Therefore, it is necessary to combine anti-PD-1/PD-L1 inhibitors with conventional treatments, including chemotherapy, radiation and targeted therapy to reduce the resistance and improve the anti-tumor effect (Figure 6).

### 4.1. Combination with Chemotherapy

ICIs, along with chemotherapy, are currently the predominant combinatory anti-tumor therapy. Chemotherapeutic drugs can augment systemic tumor-specific immune responses by enhancing neoantigen release and presentation, stimulating DC production and maturation and inducing macrophages to secrete pro-inflammatory cytokines. In addition, chemotherapy can eliminate immunosuppressive MDSC and T_reg_, which increases the susceptibility of tumor cells to cytotoxic T cells and improves the efficacy of immunotherapy [164]. The efficacy of atezolizumab in combination with albumin-bound paclitaxel versus albumin-bound paclitaxel alone has been assessed in the IMpassion130 clinical trial that enrolled 902 advanced TNBC patients. According to the interim analysis, PFS improved by 1.7 months in the combination treatment group compared to the alone treatment group in the intention-to-treat (ITT) population and by 2.5 months in the PD-L1-positive subgroup [165,166]. The phase III clinical trial KEYNOTE-355 investigated the efficacy of pembrolizumab in combination with chemotherapy (paclitaxel, paclitaxel or gemcitabine plus carboplatin) for advanced TNBC. The median PFS was higher in the combination therapy group than in the chemotherapy group, including ITT, PD-L1 positive (CPS ≥ 1) and PD-L1 high expression subgroups (CPS ≥ 10) [167]. Therefore, combining the PD-L1 inhibitors and chemotherapy may have better efficacy in TNBC.

### 4.2. Combination with Radiotherapy

Radiation can remodel the immunological context of the TME by enhancing neoantigen expression and release, triggering the release of pro-inflammatory factors and increasing tumor-infiltrating immune cells [168]. With enhanced tumor-specific antigen release and elevated antigen presentation by DCs, irradiated tumors provide an ‘in situ tumor vaccine’ for the activation of tumor-specific T cells. In addition, radiotherapy-induced apoptotic tumor cells are phagocytosed by DCs and other APCs, the antigens of which can be presented by MHC-Ⅰ molecules and activate endogenous CD8^+^ T cells [169]. The ability of radiation to promote anti-tumor T cell activation is attracting clinical attention as ICI treatment progresses, and it has been demonstrated that it can overcome ICI resistance in mouse models [169]. In a study examining the combined effects of radiation and PD-L1 inhibitors in mouse models of breast and colon cancer, it has been found that combination therapy can effectively elicit CD8^+^ T cell responses, optimize the tumor immune microenvironment and control tumor growth [170]. Radiotherapy stimulates the release of CXCL16 from mouse breast cancer cells, which recruits CXCR-6-expressing T cells into TME. In a mouse model inoculated with 4T1 cells, radiotherapy combined with PD-1 antibody delayed tumor growth and inhibited the formation of lung metastases, resulting in increased survival [171,172]. In phase II clinical trial (TONIC), radiation treatment increased TIL and CD8^+^ T cells in the TME of TNBC, making the TME “hotter” and more susceptible to PD-1 inhibitors [173]. In another phase II clinical trial (Simon 2), an ORR of 33% has been achieved with pembrolizumab in combination with radiotherapy, compared to an ORR of 18.5% with monotherapy [174]. Radiotherapy combined with immunotherapy can enhance and sustain anti-tumor immune responses for both primary and metastatic tumors, but more research is needed to determine the best radiotherapy dose and fractionated irradiation modalities for triggering systemic anti-tumor immunity.

### 4.3. The Synergistic Effect of ICIs

Immune checkpoints, like CTLA-4 and PD-1, exert their functions in distinct ways and complement each other in the control of adaptive immune responses. PD-1 induces the exhaustion of peripheral T cells, whereas CTLA-4 mainly inhibits T cell activation in the early stages. Therefore, anti-CTLA-4 antibodies can enhance the therapeutic effects of anti-PD-1 antibodies [175,176]. The immunoreceptor TIGIT is a promising new target for cancer immunotherapy. By binding to CD155 on DCs, TIGIT triggers a signaling cascade response that decreases production and secretion of IL-12 and IL-10, contributing to the formation of immune-tolerant DCs [177]. In addition, TIGIT inhibits NK cell degranulation, cytokine production and NK cell-mediated cytotoxicity in CD155^+^ tumor cells. Interaction of TIGIT^+^ NK cells with MDSCs expressing CD155 reduces the phosphorylation of ZAP70/Syk and ERK1/2, decreasing the cytolytic capacity of NK cells [178]. The TIGIT/CD155 axis has been shown to mediate resistance to ICI, which may make TIGIT blockers potential candidates for the treatment of patients with immune resistance [179,180]. Given the regulatory role of the TIGIT pathway in T cell and NK cell-mediated tumor recognition, dual blockade of PD-1 and TIGIT effectively increases the expansion of tumor antigen-specific CD8^+^ T cells in vitro, promoting tumor rejection in a mouse model [181]. In mice injected with EMT6 breast cancer cells, it has been observed that the simultaneous blockade of TIGIT and PD-1 induces stronger anti-tumor immune effects and achieves a complete response (CR) [110]. In addition, there are combination therapies with LAG-3 inhibitors and IDO inhibitors, which are believed to bring new options for patients who are resistant to PD-1/PD-L1 inhibitors [182,183]. In phase I/II clinical trial (NCT02460224), LAG525 (an antibody against LAG-3) in combination with spartalizumab (an antibody against PD-1) has shown durable responses in solid tumors, including TNBC [184]. Combining IDO inhibitor Navoximod with an anti-PD-L1 monoclonal antibody more effectively activates CTLs cells in tumors and inhibits tumor growth [185]. A phase Ib clinical trial including 66 patients with different types of malignancies (including TNBC) showed that navoximod combined with atezolizumab upregulated the expression of IDO and PD-L1 on the surface of all types of tumor cells leading to partial remission in 9% of patients and stable disease in 17%, while PD-L1^+^ patients had a slightly higher response rate than PD-L1^−^ patients [186].

### 4.4. Combination with Targeted Therapy

The TNBC with homologous recombination repair deficiency (HRD) is sensitive to PARP inhibitors, which leads to the accumulation of DNA single-strand breaks and double-strand breaks that need to be repaired by homologous recombination enzyme [187,188,189]. The application of PARP inhibitors in the context of BRCA mutations results in a synthetic lethal effect on TNBC cells via inhibiting DNA repair [190,191]. Notably, in a BRCA-mutant basal breast cancer exceptional long-term survivor, a striking tumor eradication was accompanied by a marked infiltration of immune cells containing CD8^+^ effector cells after PARP inhibitors [192]. The PARP inhibitor olaparib induces CD8^+^ T cell infiltration and activation by inducing the cGAS/STING-dependent pro-inflammatory cytokine production in tumor cells, providing a rationale for combining PARP inhibition with immunotherapies for the treatment of TNBC [193]. A phase I/II clinical trial (MEDIOLA trial) evaluated the efficacy of the PARP inhibitor Olaparib in combination with Durvalumab for the treatment of germline BRCA mutated metastatic breast cancer. The results showed that the median duration of remission was better in the 10 TNBC patients who received ICIs in combination with targeted therapy than with Olaparib alone [194]. Other clinical trials have evaluated the efficacy and safety of the PARP inhibitor Niraparib in combination with Pembrolizumab in patients with advanced TNBC. The combination therapy improved the ORR and DCR in patients with BRCA mutations compared to wild-type BRCA, resulting in significantly longer median PFS with tolerable overall adverse effects [195].

It has been found that the cyclin D-CDK4 complex reduces the stability of the PD-L1 protein. There is a consensus that the sensitivity of tumor cells to ICIs is related to the level of PD-L1 expression, thus, increasing the level and stability of PD-L1 in tumor cells may improve the therapeutic effect of ICIs. The cyclin D-CDK4/6 increases the level of transcription factor E2F and phosphorylation of the retinoblastoma (Rb), thereby triggering the cell cycle from pre-DNA synthesis (G1 phase) to DNA replication (S1 phase) [196]. The normal expression of the Rb gene is key to the effectiveness of CDK4/6 inhibitors in treating breast cancer. The study confirmed that Rb^+^ is more common in TNBC patients lacking BRCA1 mutation and the expression level of androgen receptor (AR) is positively correlated with the level of Rb expression, therefore, TNBC patients expressing Rb gene or AR^+^ TNBC patients may be sensitive to CDK4/6 inhibitors [197]. This might be because the cancer cells lacking the Rb gene could express PD-L1 at higher levels. The CDK inhibitor Palbociclib in combination with the PD-1 monoclonal antibody regimen significantly delays tumor progression and improves OS in TNBC patients [198]. Abemaciclib, a selective CDK4/6 inhibitor, reduces tumor cell growth primarily through an Rb gene-dependent mechanism, and clinical trials of abemaciclib in combination with pembrolizumab for TNBC are ongoing [199].

The receptor tyrosine kinase EGFR is overexpressed in 36%-89% of TNBC patients, which promotes cell proliferation, differentiation and migration [200,201,202,203]. In the early stage of breast cancer, EGFR overexpression is associated with reduced OS and DFS [204]. The EGFR inhibitor gefitinib enhances the efficacy of PD-1 monoclonal antibody by reducing the interaction between PD-L1 and PD-1, enhancing IL-2 expression in T cells, promoting the activation of CD8^+^ TILs and improving T cell-mediated tumor cell killing [205]. Consequently, gefitinib can reduce the survival rate of EGFR overexpressing cancer cells by reducing PD-L1 expression [206]. The potential of EGFR inhibitors for inducing anti-tumor immunity offers the feasibility of combining the EGFR-targeting therapies, including TKIs and CAR-T with ICIs in EGFR-overexpressed TNBC [207,208].

TNBCs with high levels of vascular endothelial growth factor (VEGF), a proangiogenic molecule produced by the tumors, are associated with a high risk of metastasis [209]). In the tumor immune microenvironment, VEGF-A can upregulate the expression of PD-1 and other immune checkpoints in CD8^+^ T cells, resulting in PD-1 inhibitor resistance [210]. While blocking the VEGF pathway can promote antigen-specific T cell migration, enhancing the effect of PD-L1 inhibitors like Atezolizumab [211]. Although there are few studies on TNBC, a combination of anti-angiogenic molecules with immunomodulators of inhibitory checkpoints may be a potential strategy for VEGF- producing TNBC.

### 4.5. Oncolytic Viruses

Oncolytic viruses are a class of viruses that have the ability of self-replication and can take advantage of the inactivation or defect of oncogenes in tumor cells, thus replicating in large numbers in tumor cells and causing lysis and death of tumor cells but normal cell death [212]. In the mouse in situ TNBC model and secondary transplantation tumor model, Maraba virus was found to not only have a strong anti-tumor effect on in situ TNBC but also prevent a recurrence. Systemic intravenous injection of Maraba virus was more effective than intratumoral injection [213]. Maraba virus not only effectively recruited immune cells through chemokines but also upregulated PD-L1 expression in breast cancer cells in 3 TNBC models, including 4T1, EMT6 and E0771 [214]. These results suggest that the efficacy of PD-1 inhibitors could be enhanced by oncolytic virus therapy.

### 4.6. Neoantigen-Based Immunotherapy

Recent advances in tumor immunotherapies, particularly neoantigen-based cancer vaccines and antibody-based therapies, have been widely studied in breast cancer due to their anti-tumor effects by stimulating an immune response [215]. The combination of ICIs and tumor vaccines will produce a stronger anti-tumor immune response by accelerating the initiation and activation of T cells and blocking the immunosuppressive pathway [216]. Currently, the combination of neoantigen-based tumor vaccines and ICIs are being studied in clinical trials, such as the Durvalumab and PVX-410 vaccine for stage II and III TNBC patients (NCT02826434) [217], and the Durvalumab and Vigil vaccine for metastatic breast cancer (NCT02725489) [218]. Bispecific antibodies are artificial antibodies containing two specific antigen-binding sites, which build a bridge between target cells and functional molecules to stimulate a directed immune response [219]. Due to their ability to target two different checkpoint molecules, bispecific antibodies provide a novel approach to delivering dual-drug ICIs in a single medication. These bispecific antibodies are predicted to be able to prevent the development of therapeutic resistance to existing immunotherapies [220]. Tebotelimab (MGD013) is such a bispecific tetravalent dual affinity re-targeting molecule (DART) that is designed to bind PD-1 and LAG-3. Tebotelimab is being tested in solid tumors and hematologic malignancies in combination with margetuximab, a monoclonal antibody against the Fc segment of HER2 [219,221]. According to preclinical studies, margetuximab exposure in the presence of tebotelimab leads to the upregulation of PD-1 and LAG-3, and enhances the lysis activity of immune cells. Early results from the clinical trial (NCT0321926) showed a tolerable safety profile and encouraging evidence of anti-tumor activity in patients with advanced HER2-positive cancers, including breast cancer, and preliminary evidence of clinical viability [222].

## 5. Conclusions and Perspective

The appearance of immunotherapy has led to a paradigm shift in the treatment of numerous tumors, including breast cancer. ICIs, represented by PD-1/PD-L1 inhibitors, can effectively improve the prognosis of TNBC patients by overcoming the immunosuppressive TME and restoring the recognition and killing effects of immune cells on cancer cells. Clinical trials with PD-1/PD-L1 inhibitors in TNBC are currently being conducted, however, while promising results have been achieved, several pressing issues have also been revealed. The benefit of ICI monotherapy is limited, with ORRs ranging from 5% to 23%, and even many PD-L1-positive populations do not respond to it. We discussed the resistance mechanisms to ICI treatment, including altered expression and presentation of tumor antigens, synergistic effects of multiple immune checkpoints, the suppressive effect of TME and aberrant activation of oncogenic signaling in tumor cells.

Additionally, multiple mechanisms intertwine and crosstalk in the population, with dynamic counterbalances that affect the physiological and pathological regulation of tumors and immune cells. Therefore, it is not clinically meaningful to find and modulate the resistance targets merely, and even the resistance can be re-established quickly, which cannot completely solve the ICI resistance situation. Screening people who can benefit from ICI therapy based on the predictors, such as PD-L1 expression, TIL assessment and TMB detection, is also one of the means to address the current drug resistance situation. More importantly, the combination of chemotherapy, radiotherapy and targeted therapy will also improve the response rate of TNBC patients to ICIs, but their effectiveness and safety need to be confirmed. In addition, we need to focus on how to minimize the incidence of immune-related adverse events, how to maximize the efficacy and safety of ICI, the dose tolerated by patients and the optimal duration of dosing, all of which need to be addressed by conducting more clinical trials.

## Figures and Tables

**Figure 1 cancers-15-00104-f001:**
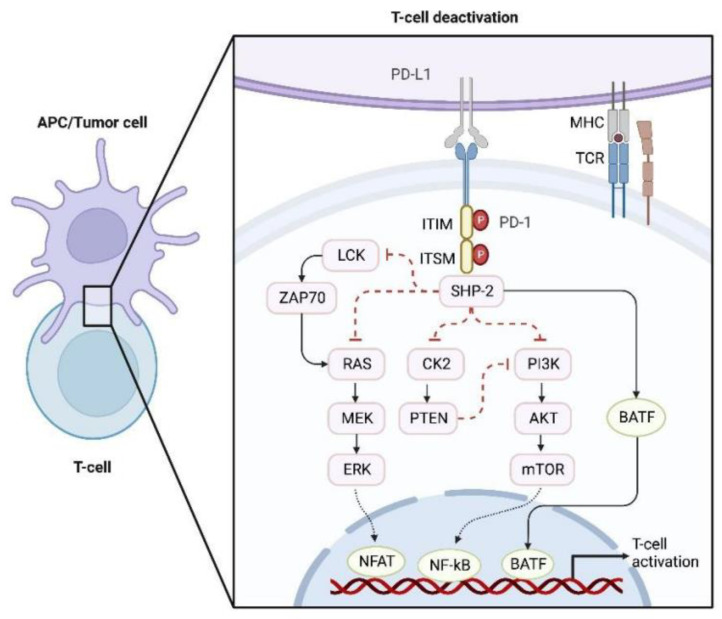
PD-1/PD-L1-mediated inhibition of T cell activation. The binding of PD-1 to PD-L1 recruits SHP-2, thereby weakening the TCR signaling pathway mediated by LCK and inhibiting the RAS-MEK-ERK and PI3K-Akt-mTOR pathways. In addition, PD-1 activation induces the expression of BTAF, which inhibits the expression of effectors for T cell activation. Collectively, the activation of T cells can be inhibited by PD-1/PD-L1-mediated inhibiting of these signaling pathways. Abbreviations: PD-1, programmed cell death protein 1; PD-L1, programmed cell death 1 ligand 1; TCR, T cell receptor; MHC, major histocompatibility complex; APC, antigen-presenting cell; ITIM, immunoreceptor tyrosine inhibitory motif; ITSM, immunoreceptor tyrosine-based switch motif; P, phosphorylation; LCK, lymphocyte-specific protein-tyrosine kinase; ZAP70, zeta chain of T cell receptor associated protein kinase 70; SHP-2, src homology-2 domain-containing protein tyrosine phosphatase; BATF, basic leucine zipper transcriptional factor ATF-like; RAS, rat sarcoma; MEK, mitogen-activated extracellular signal-regulated kinase; ERK, extracellular regulated protein kinase; NFAT, nuclear factor of activated T cell; CK2, casein kinase II; PTEN, phosphatase and tensin homolog; PI3K, phosphoinositide 3-kinase; AKT, protein kinase B; mTOR, mammalian target of rapamycin; NF-κB, nuclear factor kappa-B.

**Figure 2 cancers-15-00104-f002:**
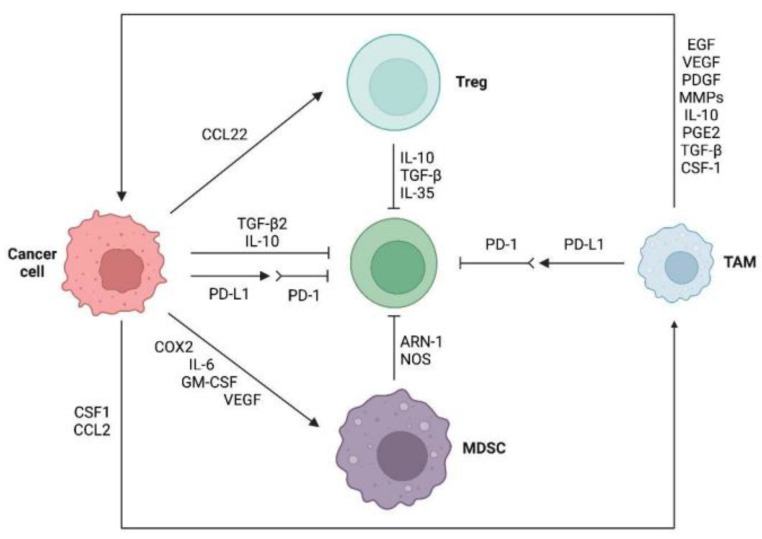
Immunosuppressive tumor microenvironment in anti-PD-1/PD-L1 therapy. TNBC cells can directly inhibit the activity of effector T cells by upregulating PD-L1 and releasing TGF-β2 and IL-10. Additionally, immunosuppressive cells, including TAM, Tregs and MDSCs, are recruited to the tumor microenvironment, where they can inhibit the anti-tumor of T cells. TAM can also promote tumor cell proliferation by secreting EGF and PDGF, facilitates tumor invasion and metastasis by releasing pro-tumor cell metastasis factors such as MMPs, triggers immune escape of tumor cells by producing IL-10, PGE2, TGF-β and CSF-1, as well as contributes to tumor microvascular growth by expressing VEGF. Abbreviations: PD-1, Programmed cell death-1; PD-L1, Programmed cell death-ligand-1; CSF1, colony stimulating factor 1; CCL2, C-C motif ligand 2; CCL22, C-C motif ligand 22; TGF-β, transforming growth factor-β; IL-6, interleukin-6; IL-10, interleukin-10; IL-35, interleukin-35; NOS, nitric oxide synthase; EGF, endothelial growth factor; VEGF, vascular endothelial growth factor; PDGF, platelet derived growth factor; MMPs, matrix metallopeptidase; PGE2, Prostaglandin E2; MDSC, myeloid-derived suppressor cell; TAM, tumor-associated macrophage.

**Figure 3 cancers-15-00104-f003:**
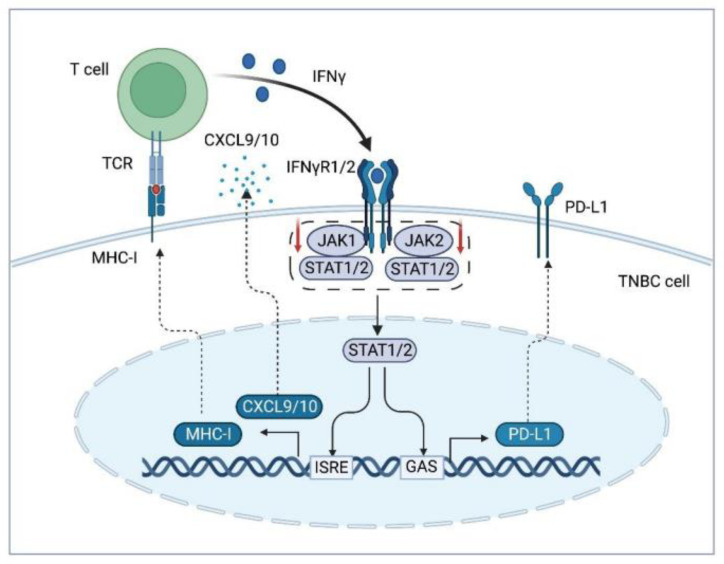
Disturbed IFN-γ signaling pathway results in resistance to anti-PD-1/PD-L1 therapy. IFN-γ secreted by T cells binds to IFNγR1/2 on the surface of TNBC cells, activating the JAK1/1-STAT1/2 pathway. The phosphorylated STAT1/2 translocates to the nucleus where it binds to GAS and ISRE, promoting the expression of MHC-II, PD-L1 and CXCL9/10. After receiving immunotherapy, tumor cells may downregulate or alter the IFN-γ signaling pathway, such as by reducing JAK1/2 activity to escape the anti-tumor effects of T cell-derived IFN-γ. Abbreviations: PD-L1, programmed cell death 1 ligand 1; TCR, T cell receptor; MHC-Ⅰ, major histocompatibility complex class Ⅰ; IFN-γ, interferon-γ; IFNγR 1/2, interferon-γ receptor 1/2; JAK 1/2, Janus kinase 1/2; STAT 1/2, signal transducer and activator of transcription 1/2; GAS, growth arrest-specific protein; ISRE, interferon-sensitive response element; CXCL9/10, C-X-C motif chemokine 9/10; TNBC, triple negative breast cancer.

**Figure 4 cancers-15-00104-f004:**
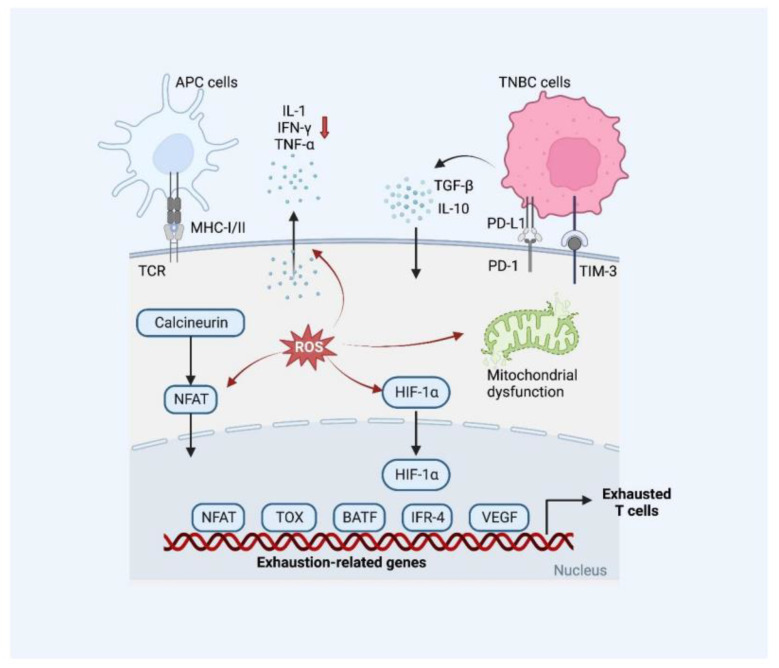
T cell exhaustion contributes to therapeutic resistance of TNBC to PD-1/PD-L1 inhibitors. Continuous exposure of T cells to tumor antigens and suppressive cytokines (IL-10, TGF-β) leads to T cell exhaustion, which results in a poor response to immunotherapy and therapeutic resistance. The accumulated ROS, increased HIF-α and activated calcium-calcineurin-NFAT signaling pathway play a key role in T cell exhaustion, which contributes to therapeutic resistance. Abbreviations: PD-1, programmed cell death protein 1; PD-L1, programmed cell death 1 ligand 1; TCR, T cell receptor; MHC, major histocompatibility complex; APC, antigen-presenting cell; TNBC, triple negative breast cancer; TIM-3, T cell immunoglobulin domain and mucin domain-3; TGF-β, transforming growth factor-beta; IL-10, interleukin-10; IL-1, interleukin-1; IFN-γ, interferon-γ; TNF-α, tumor necrosis factor-α; ROS, reactive oxygen species; NFAT, nuclear factor of activated T cells; HIF-1α, hypoxia inducible factor-1α; TOX, thymocyte selection associated high mobility group box; BATF, basic leucine zipper transcriptional factor ATF-like; IFR-4, interferon regulatory factor 4.

**Figure 5 cancers-15-00104-f005:**
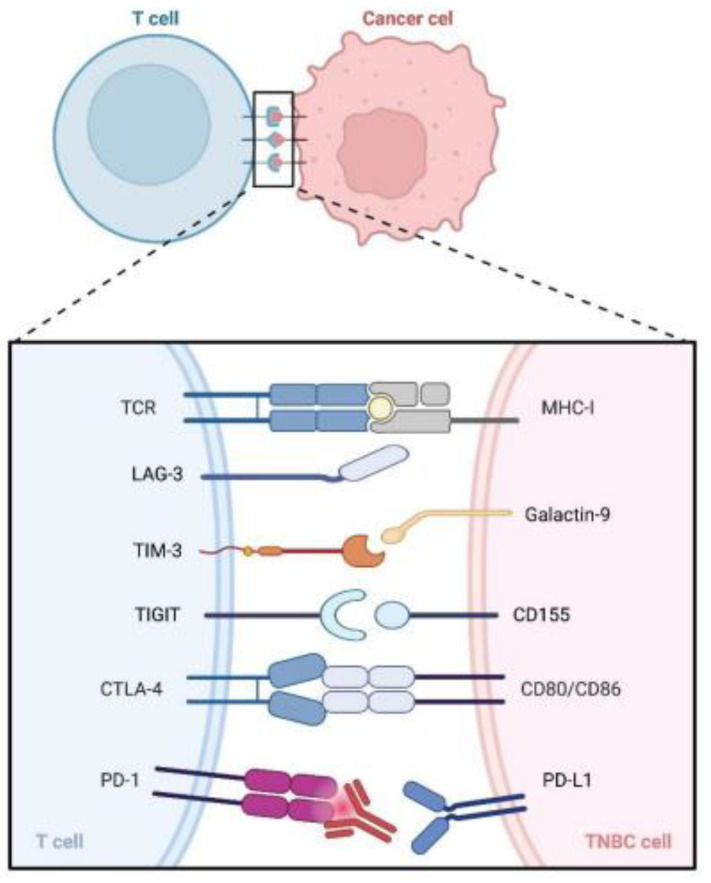
Inhibitory immune checkpoints for TNBC. TNBC cells express immune checkpoint ligands, which mediate resistance to PD-1/PD-L1 inhibitors by inhibiting T cell function. After the PD-1/PD-L1 signaling pathway is blocked by inhibitors, other immune checkpoints, such as CTLA-4, TIGIT, TIM-3 and LAG-3, are compensated upregulated. Abbreviations: PD-1, programmed cell death protein 1; PD-L1, programmed cell death 1 ligand 1; CTLA-4, cytotoxic T-lymphocyte-associated protein 4; TIGIT, T cell immunoglobulin and ITIM domain; TIM-3, T cell immunoglobulin domain and mucin domain-3; LAG-3, lymphocyte activation gene-3; TCR, T cell receptor; MHC-II, major histocompatibility complex class II; TNBC, triple negative breast cancer.

**Figure 6 cancers-15-00104-f006:**
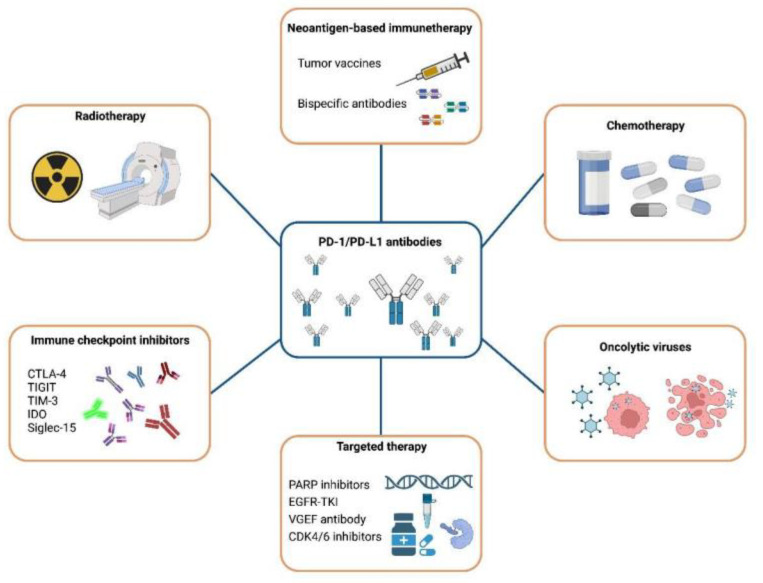
Combination therapies with anti-PD-1/PD-L1 antibodies. Combination therapies: including PD-1/PD-L1 antibodies along with radiotherapy, chemotherapy, other immune checkpoint antibodies, targeted therapy, oncolytic viruses and neoantigen-based immunotherapy are being developed to treat cancer patients effectively. Abbreviations: PD-1, Programmed cell death-1; PD-L1, Programmed cell death-ligand-1; CTLA-4, Cytotoxic T Lymphocyte-Associated Antigen-4; TIGIT, T cell immunoglobulin and ITIM domains; ITIM, immunoreceptor tyrosine-based inhibitory motif; TIM-3, T cell immunoglobulin domain and mucin domain-3; IDO, indoleamine 2, 3 dioxygenase; Siglec-15, Sialic Acid Binding Ig Like Lectin-15; PPAP inhibitors, Poly ADP-ribosepolymerase inhibitor; EGFR-TKI, epidermal growth factor receptor-tyrosine kinase inhibitors; VEGF antibody, vascular endothelial growth factor antibody; CKD4/6 inhibitors, cyclin dependent kinase4/6 inhibitors.

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
