# Peer review of "Mechanisms and Strategies to Overcome PD-1/PD-L1 Blockade Resistance in Triple-Negative Breast Cancer"

_cancers, 2022, doi:10.3390/cancers15010104_

Round 1
Reviewer 1 Report
This paper summarized the recent study about PD-1/PD-L1 blockade resistance in triple-negative breast cancer (TNBC). Resistance is problematic in chemotherapy, and this article will help us understand the current therapeutic strategies and future research on when ICIs are used.
The literature was collected from the recent progress and enough amount. The contents are summarized from research and clinical aspects. The figures were illustrated clearly and supported the text body.
I suggest publishing this manuscript after these minor revisions to the format.
- Some font sizes for sub-chapters need to be more consistent.
- For example, 2.3 and 2.4 are smaller than 2.1 and 2.2. The same thing happened on 3.2, 3.4, and all 4.
Author Response
This paper summarized the recent study about PD-1/PD-L1 blockade resistance in triple-negative breast cancer (TNBC). Resistance is problematic in chemotherapy, and this article will help us understand the current therapeutic strategies and future research on when ICIs are used. The literature was collected from the recent progress and enough amount. The contents are summarized from research and clinical aspects. The figures were illustrated clearly and supported the text body. I suggest publishing this manuscript after these minor revisions to the format.
Some font sizes for sub-chapters need to be more consistent. For example, 2.3 and 2.4 are smaller than 2.1 and 2.2. The same thing happened on 3.2, 3.4, and all 4.
Response: Thank you very much for pointing out the mistake. We have corrected the font sizes as suggested.

Reviewer 2 Report
This is a well-written manuscript and provides a deep overview of the existing literature on a such important topic. The authors have critically analysed and evaluated new insights into recent advances. I really enjoyed reading the paper.
However, it would be helpful to clarify why PD-L1 expression did not differentiate responders from non-responders patients to immunotherapy, and PD-L1 negative patients respond to ICIs. Another point is that patients with aberrant activation of MIC are resistant to ICI therapy. MYC increases PD-L1 expression. However, the high level of PD-L1 expression has been proposed as a biomarker for predicting the ICI efficacy.
Author Response
This is a well-written manuscript and provides a deep overview of the existing literature on a such important topic. The authors have critically analysed and evaluated new insights into recent advances. I really enjoyed reading the paper.
1.However, it would be helpful to clarify why PD-L1 expression did not differentiate responders from non-responders patients to immunotherapy, and PD-L1 negative patients respond to ICIs.
Response: Thank you very much for your valuable comments. We have added a discussion on why PD-L1 does not accurately differentiate those who benefit from ICI treatment. (Lines 318-326)
2.Another point is that patients with aberrant activation of MYC are resistant to ICI therapy. MYC increases PD-L1 expression. However, the high level of PD-L1 expression has been proposed as a biomarker for predicting the ICI efficacy.
Response: Thank you very much for your kind suggestion. We have added a discussion on why aberrant activation of MYC leads to elevated PD-L1 but serves as a biomarker for poor response to ICI therapy in our revised manuscript. (Lines 383-388)
